# Handling Missing Data with Graph Representation Learning

**Jiaxuan You**[1]* **Xiaobai Ma**[2]* **Daisy Yi Ding**[3]* **Mykel Kochenderfer**[2] **Jure Leskovec**[1]

[1]Department of Computer Science, [2]Department of Aeronautics and Astronautics,
and [3]Department of Biomedical Data Science, Stanford University
{jiaxuan, jure}@cs.stanford.edu
{maxiaoba, dingd, mykel}@stanford.edu

## Abstract

Machine learning with missing data has been approached in two different ways, including *feature imputation* where missing feature values are estimated based on observed values and *label prediction* where downstream labels are learned directly from incomplete data. However, existing imputation models tend to have strong prior assumptions and cannot learn from downstream tasks, while models targeting label prediction often involve heuristics and can encounter scalability issues. Here we propose GRAPE, a graph-based framework for feature imputation as well as label prediction. GRAPE tackles the missing data problem using a *graph representation*, where the observations and features are viewed as two types of nodes in a bipartite graph, and the observed feature values as edges. Under the GRAPE framework, the *feature imputation* is formulated as an *edge-level prediction* task and the *label prediction* as a *node-level prediction* task. These tasks are then solved with Graph Neural Networks. Experimental results on nine benchmark datasets show that GRAPE yields 20% lower mean absolute error for imputation tasks and 10% lower for label prediction tasks, compared with existing state-of-the-art methods.

## 1 Introduction

Issues with learning from incomplete data arise in many domains including computational biology, clinical studies, survey research, finance, and economics [6, 32, 46, 47, 53]. The missing data problem has previously been approached in two different ways: *feature imputation* and *label prediction*. Feature imputation involves estimating missing feature values based on observed values [8, 9, 11, 14, 15, 17, 22, 34, 44, 45, 47–50, 56], and label prediction aims to directly accomplish a downstream task, such as classification or regression, with the missing values present in the input data [2, 5, 10, 15, 16, 23, 37, 40, 42, 52, 54].

Statistical methods for feature imputation often provide useful theoretical properties but exhibit notable shortcomings: (1) they tend to make strong assumptions about the data distribution; (2) they lack the flexibility for handling mixed data types that include both continuous and categorical variables; (3) matrix completion based approaches cannot generalize to unseen samples and require retraining when the model encounters new data samples [8, 9, 22, 34, 44, 47]. When it comes to models for label prediction, existing approaches such as tree-based methods rely on heuristics [5] and tend to have scalability issues. For instance, one of the most popular procedures called surrogate splitting does not scale well, because each time an original splitting variable is missing for some observation it needs to rank all other variables as surrogate candidates and select the best alternative.

Recent advances in deep learning have enabled new approaches to handle missing data. Existing imputation approaches often use deep generative models, such as Generative Adversarial Networks

---

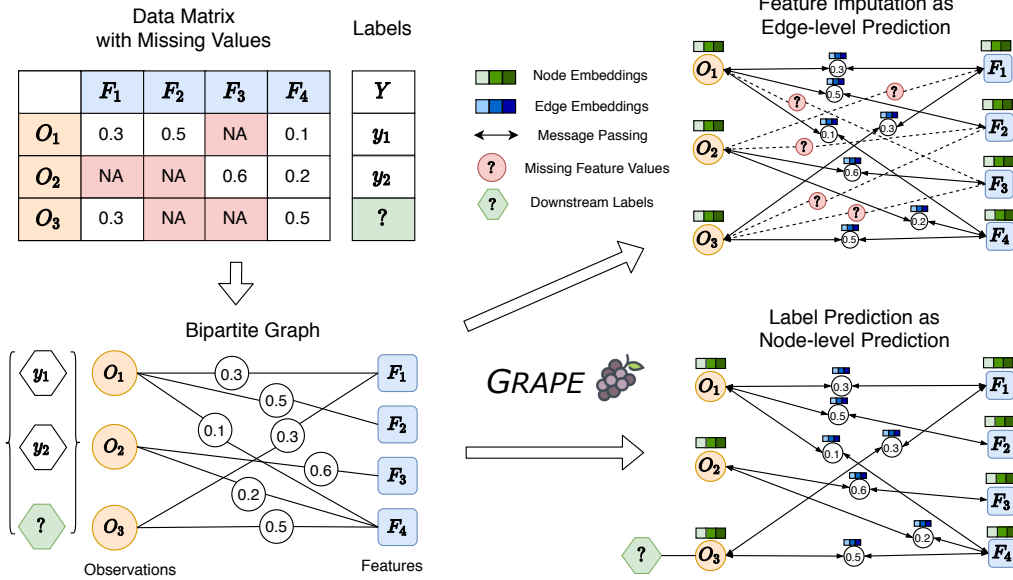

Figure 1: In the GRAPE framework, we construct a bipartite graph from the data matrix with missing feature values, where the entries of the matrix in red indicate the missing values (**Top Left**). To construct the graph, the observations $O$ and features $F$ are considered as two types of nodes and the observed values in the data matrix are viewed as weighted/attributed edges between the observation and feature nodes (**Bottom Left**). With the constructed graph, we formulate the feature imputation problem and the label prediction problem as edge-level (**Top right**) and node-level (**Bottom right**) prediction tasks, respectively. The tasks can then be solved with our GRAPE GNN model that learns node and edge embeddings through rounds of message passing.

(GANs) [56] or autoencoders [17, 50], to reconstruct missing values. While these models are flexible, they have several limitations: (1) when imputing missing feature values for a given observation, these models fail to make full use of feature values from other observations; (2) they tend to make biased assumptions about the missing values by initializing them with special default values.

Here, we propose GRAPE[1], a general framework for feature imputation and label prediction in the presence of missing data. Our key innovation is to formulate the problem using a *graph representation*, where we construct a bipartite graph with observations and features as two types of nodes, and the observed feature values as attributed edges between the observation and feature nodes (Figure 1). Under this graph representation, the *feature imputation* can then be naturally formulated as an *edge-level prediction* task, and the *label prediction* as a *node-level prediction* task.

GRAPE solves both tasks via Graph Neural Networks (GNNs). Specifically, GRAPE adopts a GNN architecture inspired by the GraphSAGE model [20], while having three innovations in its design: (1) since the edges in the graph are constructed based on the data matrix and have rich attribute information, we introduce *edge embeddings* during message passing and incorporate both discrete and continuous edge features in the message computation; (2) we design *augmented node features* to initialize observation and feature nodes, which provides greater representation power and maintains inductive learning capabilities; (3) to overcome the common issue of overfitting in the missing data problem, we employ an *edge dropout* technique that greatly boosts the performance of GRAPE.

We compare GRAPE with the state-of-the-art feature imputation and label prediction algorithms on 9 benchmark datasets from the UCI Machine Learning Repository [1]. In particular, GRAPE yields 20% lower mean absolute error (MAE) for the imputation tasks and 10% lower MAE for the prediction tasks at the 30% data missing rate. Finally, we demonstrate GRAPE's strong generalization ability by showing its superior performance on unseen observations without the need for retraining.

Overall, our approach has several important benefits: (1) by creating a bipartite graph structure we create connections between different features (via observations) and similarly between the observations (via features); (2) GNN elegantly harnesses this structure by learning to propagate and borrow information from other features/observations in a graph localized way; (3) GNN allows us to model both feature imputation as well as label prediction in an end-to-end fashion, which as we show in experiments leads to strong performance improvements.

## 2  Related Work

**Feature imputation**. Successful statistical approaches for imputation include joint modeling with Expectation-Maximization [11, 14, 15, 25], multivariate imputation by chained equations (MICE) [7, 38, 45, 48, 49], $k$-nearest neighbors (KNN) [27, 47], and matrix completion [8, 9, 22, 34, 44, 47]. However, joint modeling tends to make assumptions about the data distribution through a parametric density function; joint modeling and matrix completion lack the flexibility to handle data of mixed modalities; MICE and KNN cannot accomplish imputation while adapting to downstream tasks.

Recently, deep learning models have also been used to tackle the feature imputation problem [17, 43, 50, 56]. However, these models have important limitations. Denoising autoencoder (DAE) models [17, 50] and GAIN [56] only use a single observation as input to impute the missing features. In contrast, GRAPE explicitly captures the complex interactions between multiple observations and features. GNN-based approaches have also been proposed in the context of matrix completion [3, 21, 35, 62, 63]. However, they often make the assumption of finite, known-range values in their model design, which limits their applicability to imputation problems with continuous values. In contrast, GRAPE can handle both continuous and discrete feature values.

**Label prediction with the presence of missing data**. Various models have been adapted for label prediction with the presence of missing data, including tree-based approaches [5, 54], probabilistic modeling [15], logistic regression [52], support vector machines [10, 37], deep learning-based models [2, 18, 42], and many others [16, 23, 30, 40]. Specifically, decision tree is a classical statistical approach that can handle missing values for the label prediction task [5]. With the surrogate splitting procedure, decision tree uses a single surrogate variable to replace the original splitting variable with missing values, which is effective but inefficient, and has been shown to be inferior to the "impute and then predict" procedure [13]. Random forests further suffer from the scalability issues as they consist of multiple decision trees [31, 54]. In contrast, GRAPE handles the missing feature entries naturally with the *graph representation* without any additional heuristics. The computation of GRAPE is efficient and easily parallelizable with modern deep learning frameworks.

**Overall discussion**. In GRAPE implementation, we adopt several successful GNN design principles. Concretely, our core architecture is inspired by GraphSAGE [20]; we apply GraphSAGE to bipartite graphs following G2SAT [59]; we use edge dropout in [39]; we use one-hot auxiliary node features which has been used in [36, 60]; we follow the GNN design guidelines in [61] to select hyperparameters. Moreover, matrix completion tasks have been formulated as bipartite graphs and solved via GNNs in [3, 62]; however, they only consider the feature imputation task with discrete feature values. We emphasize that our main contribution is *not the particular GNN model but the graph-based framework for the general missing data problem*. GRAPE is the first graph-based solution to both feature imputation and label prediction aspects of the missing data problem.

## 3  The GRAPE Framework

### 3.1  Problem Definition

Let $\mathbf{D} \in \mathbb{R}^{n \times m}$ be a feature matrix consisting of $n$ data points and $m$ features. The $j$-th feature of the $i$-th data point is denoted as $\mathbf{D}_{ij}$. In the missing data problem, certain feature values are missing, denoted as a mask matrix $\mathbf{M} \in \{0, 1\}^{n \times m}$ where the value of $\mathbf{D}_{ij}$ can be observed only if $\mathbf{M}_{ij} = 1$. Usually, datasets come with labels of a downstream task. Let $\mathbf{Y} \in \mathbb{R}^n$ be the label for a downstream task and $\mathbf{V} \in \{0, 1\}^n$ the train/test partition, where $\mathbf{Y}_i$ can be observed at training test only if $\mathbf{V}_i = 1$. We consider two tasks: (1) feature imputation, where the goal is to predict the missing feature values $\mathbf{D}_{ij}$ at $\mathbf{M}_{ij} = 0$; (2) label prediction, where the goal is to predict test labels $\mathbf{Y}_i$ at $\mathbf{V}_i = 0$.

## 3.2 Missing Data Problem as a Graph Prediction Task

The key insight of this paper is to represent the feature matrix with missing values as a *bipartite graph*. Then the feature imputation problem and the label prediction problem can naturally be formulated as node prediction and edge prediction tasks (Figure 1).

**Feature matrix as a bipartite graph**. The feature matrix $\mathbf{D}$ and the mask $\mathbf{M}$ can be represented as an undirected bipartite graph $\mathcal{G} = (\mathcal{V}, \mathcal{E})$, where $\mathcal{V}$ is the node set that consists of two types of nodes $\mathcal{V} = \mathcal{V}_D \cup \mathcal{V}_F$, $\mathcal{V}_D = \{u_1, ..., u_n\}$ and $\mathcal{V}_F = \{v_1, \ldots, v_m\}$, $\mathcal{E}$ is the edge set where edges only exist between nodes in different partitions: $\mathcal{E} = \{(u_i, v_j, \mathbf{e}_{u_i v_j}) \mid u_i \in \mathcal{V}_D, v_j \in \mathcal{V}_F, \mathbf{M}_{ij} = 1\}$, where the edge feature, $\mathbf{e}_{u_i v_j}$, takes the value of the corresponding feature $\mathbf{e}_{u_i v_j} = \mathbf{D}_{ij}$. If $\mathbf{D}_{ij}$ is a discrete variable then it is transformed to a one-hot vector then assigned to $\mathbf{e}_{u_i v_j}$. To simplify the notation $\mathbf{e}_{u_i v_j}$, we use $\mathbf{e}_{ij}$ in the context of feature matrix $\mathbf{D}$, and $\mathbf{e}_{uv}$ in the context of graph $\mathcal{G}$.

**Feature imputation as edge-level prediction**. Using the definitions above, imputing missing features can be represented as learning the edge value prediction mapping: $\hat{\mathbf{D}}_{ij} = \hat{\mathbf{e}}_{ij} = f_{ij}(\mathcal{G})$ by minimizing the difference between $\hat{\mathbf{D}}_{ij}$ and $\mathbf{D}_{ij}, \forall \mathbf{M}_{ij} = 0$. When imputing discrete attributes, we use cross entropy loss. When imputing continuous values, we use MSE loss.

**Label prediction as node-level prediction**. Predicting downstream node labels can be represented as learning the mapping: $\hat{\mathbf{Y}}_i = g_i(\mathcal{G})$ by minimizing the difference between $\hat{\mathbf{Y}}_i$ and $\mathbf{Y}_i, \forall \mathbf{V}_i = 0$.

## 3.3 Learning with GRAPE

GRAPE adopts a GNN architecture inspired by GraphSAGE [20], which is a variant of GNNs that has been shown to have strong inductive learning capabilities across different graphs. We extend GraphSAGE to a bipartite graph setting by adding multiple important components that ensure its successful application to the missing data problem.

**GRAPE GNN architecture**. Given that our bipartite graph $\mathcal{G}$ has important information on its edges, we modify GraphSAGE architecture by introducing *edge embeddings*. At each GNN layer $l$, the message passing function takes the concatenation of the embedding of the source node $\mathbf{h}_v^{(l-1)}$ and the edge embedding $\mathbf{e}_{uv}^{(l-1)}$ as the input:

$$\mathbf{n}_v^{(l)} = \text{AGG}_l\Big(\sigma(\mathbf{P}^{(l)} \cdot \text{CONCAT}(\mathbf{h}_v^{(l-1)}, \mathbf{e}_{uv}^{(l-1)}) \mid \forall u \in \mathcal{N}(v, \mathcal{E}_{drop}))\Big) \tag{1}$$

where $\text{AGG}_l$ is the aggregation function, $\sigma$ is the non-linearity, $\mathbf{P}^{(l)}$ is the trainable weight, $\mathcal{N}$ is the node neighborhood function. Node embedding $\mathbf{h}_v^{(l)}$ is then updated using:

$$\mathbf{h}_v^{(l)} = \sigma(\mathbf{Q}^{(l)} \cdot \text{CONCAT}(\mathbf{h}_v^{(l-1)}, \mathbf{n}_v^{(l)})) \tag{2}$$

where $\mathbf{Q}^{(l)}$ is the trainable weight, we additionally update the edge embedding $\mathbf{e}_{uv}^{(l)}$ by:

$$\mathbf{e}_{uv}^{(l)} = \sigma(\mathbf{W}^{(l)} \cdot \text{CONCAT}(\mathbf{e}_{uv}^{(l-1)}, \mathbf{h}_u^{(l)}, \mathbf{h}_v^{(l)})) \tag{3}$$

where $\mathbf{W}^{(l)}$ is the trainable weight. To make edge level predictions at the $L$-th layer:

$$\hat{\mathbf{D}}_{uv} = \mathbf{O}_{edge}(\text{CONCAT}(\mathbf{h}_u^{(L)}, \mathbf{h}_v^{(L)})) \tag{4}$$

The node-level prediction is made using the imputed dataset $\hat{\mathbf{D}}$:

$$\hat{\mathbf{Y}}_u = \mathbf{O}_{node}(\hat{\mathbf{D}}_{u\cdot}) \tag{5}$$

where $\mathbf{O}_{edge}$ and $\mathbf{O}_{node}$ are feedforward neural networks.

**Augmented node features for bipartite message passing**. Based on our definition, nodes in $\mathcal{V}_D$ and $\mathcal{V}_F$ do not naturally come with features. The straightforward approach would be to augment nodes with constant features. However, such formulation would make GRAPE hard to differentiate messages from different feature nodes in $\mathcal{V}_F$. In real-world applications, different features can represent drastically different semantics or modalities. For example in the *Boston Housing* dataset from UCI [1], some features are categorical such as if the house is by the Charles River, while others are continuous such as the size of the house.

**Algorithm 1** GRAPE forward computation

---

**Input:** Graph $\mathcal{G} = (\mathcal{V}; \mathcal{E})$; Number of layers $L$; Edge dropout rate $r_{drop}$; Weight matrices $\mathbf{P}^{(l)}$ for *message passing*, $\mathbf{Q}^{(l)}$ for *node updating*, and $\mathbf{W}^{(l)}$ for *edge updating*; non-linearity $\sigma$; aggregation functions $\text{AGG}_l$; neighborhood function $\mathcal{N} : v \times \mathcal{E} \to 2^{\mathcal{V}}$

**Output:** Node embeddings $\mathbf{h}_v$ corresponding to each $v \in \mathcal{V}$

1: $\mathbf{h}_v^{(0)} \leftarrow \text{INIT}(v), \forall v \in \mathcal{V}$
2: $\mathbf{e}_{uv}^{(0)} \leftarrow \mathbf{e}_{uv}, \forall \mathbf{e}_{uv} \in \mathcal{E}$
3: $\mathcal{E}_{drop} \leftarrow \text{DROPEDGE}(\mathcal{E}, r_{drop})$
4: **for** $l \in \{1, \ldots, L\}$
5:    **for** $v \in \mathcal{V}$
6:       $\mathbf{n}_v^{(l)} = \text{AGG}_l\Big(\sigma(\mathbf{P}^{(l)} \cdot \text{CONCAT}(\mathbf{h}_v^{(l-1)}, \mathbf{e}_{uv}^{(l-1)}) \mid \forall u \in \mathcal{N}(v, \mathcal{E}_{drop}))\Big)$
7:       $\mathbf{h}_v^{(l)} = \sigma(\mathbf{Q}^{(l)} \cdot \text{CONCAT}(\mathbf{h}_v^{(l-1)}, \mathbf{n}_v^{(l)}))$
8:    **for** $(u, v) \in \mathcal{E}_{drop}$
9:       $\mathbf{e}_{uv}^{(l)} = \sigma(\mathbf{W}^{(l)} \cdot \text{CONCAT}(\mathbf{e}_{uv}^{(l-1)}, \mathbf{h}_u^{(l)}, \mathbf{h}_v^{(l)}))$
10:   $z_v \leftarrow h_v^L$

---

Instead, we propose to use $m$-dimensional one-hot node features for each node in $\mathcal{V}_F$ ($m = |\mathcal{V}_F|$), while using $m$-dimensional[1] constant vectors as node feature for data nodes in $\mathcal{V}_F$:

$$\text{INIT}(v) = \begin{cases} \mathbf{1} & v \in \mathcal{V}_D \\ \text{ONEHOT} & v \in \mathcal{V}_F \end{cases} \tag{6}$$

Such a formulation leads to a better representational power to differentiate feature nodes with different underlying semantics or modalities. Additionally, the formulation has the capability of generalizing the trained GRAPE to completely unseen data points in the given dataset. Furthermore, it allows us to transfer knowledge from an external dataset with the same set of features to the dataset of interest, which is particularly useful when the external dataset provides rich information on the interaction between observations and features (as captured by GRAPE). For example, as a real-world application in biomedicine, gene expression data can be used to predict disease types and frequently contain missing values. If we aim to impute missing values in a gene expression dataset of a small cohort of lung cancer patients, public datasets, e.g., the Cancer Genome Atlas Program (TCGA) [51] can be first leveraged to train GRAPE, where rich interactions between patients and features are learned. Then, the trained GRAPE can be applied to our smaller dataset of interest to accomplish imputation.

**Improved model generalization with edge dropout**. When doing feature imputation, a naive way of training GRAPE is to directly feed $\mathcal{G} = (\mathcal{V}; \mathcal{E})$ as the input. However, since all the observed edge values are used as the input, an identity mapping $\hat{\mathbf{D}}_{ij} = \mathbf{e}_{ij}^{(0)}$ is enough to minimize the training loss; therefore, GRAPE trained under this setting easily overfits the training set. To force the model to generalize to unseen edge values, we randomly mask out edges $\mathcal{E}$ with dropout rate $r_{drop}$:

$$\text{DROPEDGE}(\mathcal{E}, r_{drop}) = \{(u_i, v_j, _{ij}) \mid (u_i, v_j, \mathbf{e}_{ij}) \in \mathcal{E}, \mathbf{M}_{drop,ij} > r_{drop}\} \tag{7}$$

where $\mathbf{M}_{drop} \in \mathbb{R}^{n \times m}$ is a random matrix sampled uniformly in $(0, 1)$. This approach is similar to DropEdge [39], but with a more direct motivation for feature imputation. At test time, we feed the full graph $\mathcal{G}$ to GRAPE. Overall, the complete computation of GRAPE is summarized in Algorithm 1.

## 4 Experiments

### 4.1 Experimental Setup

**Datasets**. We conduct experiments on 9 datasets from the UCI Machine Learning Repository [1]. The datasets come from different domains including civil engineering (CONCRETE, ENERGY), biology (PROTEIN), thermal dynamics (NAVAL), etc. The smallest dataset (YACHT) has 314 observations and 6 features, while the largest dataset (PROTEIN) has over 45,000 observations and 9 features. The datasets are fully observed; therefore, we introduce missing values by randomly removing values in the data matrix. The attribute values are scaled to $[0, 1]$ with a MinMax scaler [29].

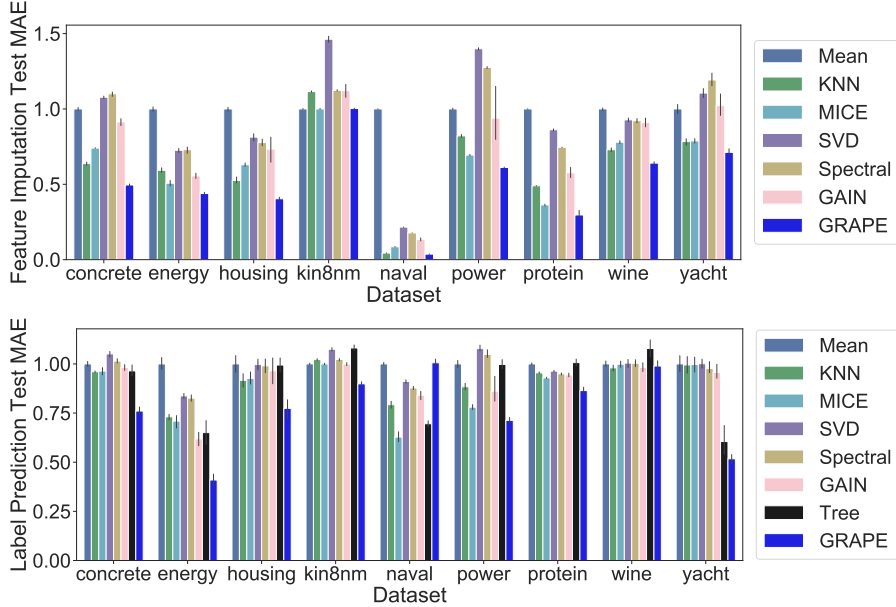

Figure 2: Averaged MAE of *feature imputation* (upper) and *label prediction* (lower) on UCI datasets over 5 trials at data missing level of 0.3. The result is normalized by the average performance of Mean imputation. GRAPE yields 20% lower MAE for imputation and 10% lower MAE for prediction compared with the best baselines (KNN for imputation and MICE for prediction).

**Baseline models**. We compare our model against five commonly used imputation methods. We also compare with a state-of-the-art deep learning based imputation model as well as a decision tree based label prediction model. More details on the baseline models are provided in the Appendix.

- Mean imputation (Mean): The method imputes the missing $\mathbf{D}_{ij}$ with the mean of all the samples with observed values in dimension $j$.
- K-nearest neighbors (KNN): The method imputes the missing value $\mathbf{D}_{ij}$ using the KNNs that have observed values in dimension $j$ with weights based on the Euclidean distance to sample $i$.
- Multivariate imputation by chained equations (MICE): The method runs multiple regression where each missing value is modeled conditioned on the observed non-missing values.
- Iterative SVD (SVD) [47]: The method imputes missing values based on matrix completion with iterative low-rank SVD decomposition.
- Spectral regularization algorithm (Spectral) [34]: This matrix completion model uses the nuclear norm as a regularizer and imputes missing values with iterative soft-thresholded SVD.
- GAIN [56], state-of-the-art deep imputation model with generative adversarial training [19].
- Decision tree (Tree) [5], a commonly used statistical method that can handle missing values for label prediction. We consider this baseline only for the label prediction task.[1]

**GRAPE configurations**. For all experiments, we train GRAPE for 20,000 epochs using the Adam optimizer [28] with a learning rate at 0.001. For all *feature imputation* tasks, we use a 3-layer GNN with 64 hidden units and RELU activation. The $\text{AGG}_l$ is implemented as a mean pooling function $\text{MEAN}(\cdot)$ and $\mathbf{O}_{edge}$ as a multi-layer perceptron (MLP) with 64 hidden units. For *label prediction* tasks, we use two GNN layers with 16 hidden units. $\mathbf{O}_{edge}$ and $\mathbf{O}_{node}$ are implemented as linear layers. The edge dropout rate is set to $r_{drop} = 0.3$. For all experiments, we run 5 trials with different random seeds and report the mean and standard deviation of the results.

## 4.2 Feature Imputation

**Setup**. We first compare the feature imputation performance of GRAPE and all other imputation baselines. Given a full data matrix $\mathbf{D} \in \mathbb{R}^{n \times m}$, we generate a random mask matrix $\mathbf{M} \in \{0, 1\}^{n \times m}$

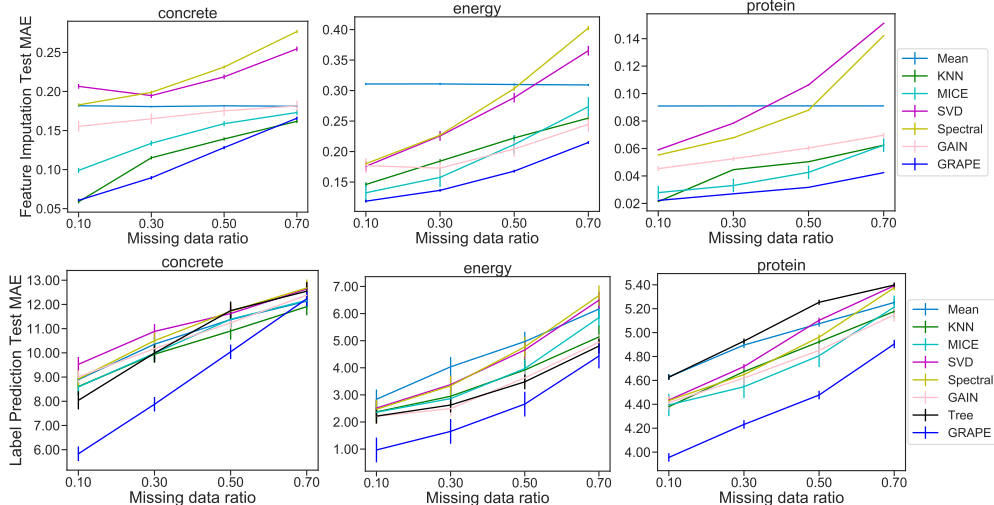

Figure 3: Averaged MAE of *feature imputation* (upper) and *label prediction* (lower) with *different missing ratios* over 5 trials. GRAPE yields 12% lower MAE on imputation and 2% lower MAE on prediction tasks across different missing data ratios.

with $P(\mathbf{M}_{ij} = 0) = r_{miss}$ at a data missing level $r_{miss} = 0.3$. A bipartite graph $\mathcal{G} = (\mathcal{V}, \mathcal{E})$ is then constructed based on $\mathbf{D}$ and $\mathbf{M}$ as described in Section 3.2. $\mathcal{G}$ is used as the input to GRAPE at both the training and test time. The training loss is defined as the mean squared error (MSE) between $\mathbf{D}_{ij}$ and $\hat{\mathbf{D}}_{ij}, \forall \mathbf{M}_{ij} = 1$. The test metric is defined as the mean absolute error (MAE) between $\mathbf{D}_{ij}$ and $\hat{\mathbf{D}}_{ij}, \forall \mathbf{M}_{ij} = 0$.

**Results**. As shown in Figure 2, GRAPE has the lowest MAE on all datasets and its average error is 20% lower compared with the best baseline (KNN). Since there are significant differences between the characteristics of different datasets, statistical methods often need to adjust its hyper-parameters accordingly, such as the cluster number in KNN, the rank in SVD, and the sparsity in Spectral. On the contrary, GRAPE is able to adjust its trainable parameters adaptively through loss backpropagation and learn different observation-feature relations for different datasets. Compared with GAIN, which uses an MLP as the generative model, the GNN used in GRAPE is able to explicitly model the information propagation process for predicting missing feature values.

## 4.3 Label Prediction

**Setup**. For label prediction experiments, with the same input graph $\mathcal{G}$, we have an additional label vector $\mathbf{Y} \in \mathbb{R}^n$. We randomly split the labels $\mathbf{Y}$ into 70/30% training and test sets, $\mathbf{Y}_{train}$ and $\mathbf{Y}_{test}$ respectively. The training loss is defined as the MSE between the true $\mathbf{Y}_{train}$ and the predicted $\hat{\mathbf{Y}}_{train}$. The test metric is calculated based on the MAE between $\mathbf{Y}_{test}$ and $\hat{\mathbf{Y}}_{test}$. For baselines except decision tree, since no end-to-end approach is available, we first impute the data and then do linear regression on the imputed data matrix for predicting $\hat{\mathbf{Y}}$.

**Results**. As is shown in Figure 2, on all datasets except NAVAL and WINE, GRAPE has the best performance. On WINE dataset, all methods have comparable performance. The fact that the performance of all methods are close to the Mean method indicates that the relation between the labels and observations in WINE is relatively simple. For the dataset NAVAL, the imputation errors of all models are very small (both relative to Mean and on absolute value). In this case, a linear regression on the imputed data is enough for label prediction. Across all datasets, GRAPE yields 10% lower MAE compared with best baselines. The improvement of GRAPE could be explained by two reasons: first, the better handling of missing data with GRAPE where the known information and the missing values are naturally embedded in the graph; and second, the end-to-end training.

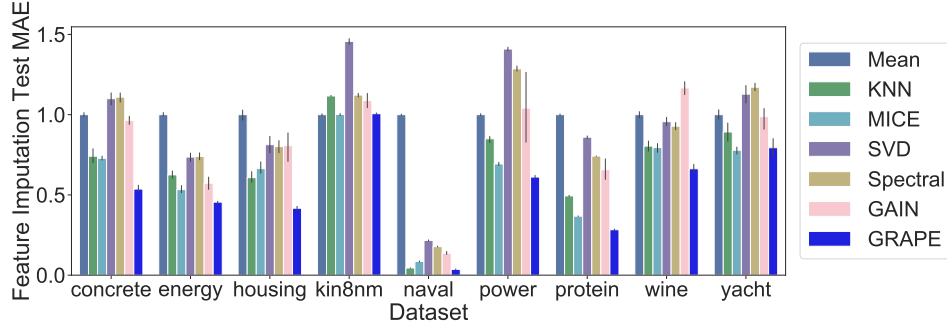

Figure 4: Averaged MAE of *feature imputation on unseen data* in UCI datasets over 5 trials. The result is normalized by the average performance of Mean imputation. GRAPE yields 21% lower MAE compared with best baselines (MICE).

## 4.4 Robustness against Different Data Missing Levels

**Setup**. To examine the robustness of GRAPE with respect to the missing level of the data matrix. We conduct the same experiments as in Sections 4.2 and 4.3 with different missing levels of $r_{miss} \in \{0.1, 0.3, 0.5, 0.7\}$.

**Results**. The curves in Figure 3 demonstrate the performance change of all methods as the missing ratio increases. GRAPE yields -8%, 20%, 20%, and 17% lower MAE on imputation tasks, and -15%, 10%, 10%, and 4% lower MAE on prediction tasks across all datasets over missing ratios of 0.1, 0.3, 0.5, and 0.7, respectively. In missing ratio of 0.1, the only baseline that behaves better than GRAPE is KNN. As in this case, the known information is adequate for the nearest-neighbor method to make good predictions. As the missing ratio increases, the prediction becomes harder and the GRAPE's ability to coherently combine all known information becomes more important.

## 4.5 Generalization on New Observations

**Setup**. We further investigate the *generalization* ability of GRAPE. Concretely, we examine whether a trained GRAPE can be successfully applied to new observations that are not in the training dataset. A good generalization ability reduces the effort of re-training when there are new observations being recorded after the model is trained. We randomly divide the $n$ observations in $\mathbf{D} \in \mathbb{R}^{n \times m}$ into two sets, represented as $\mathbf{D}_{train} \in \mathbb{R}^{n_{train} \times m}$ and $\mathbf{D}_{test} \in \mathbb{R}^{n_{test} \times m}$, where $\mathbf{D}_{train}$ and $\mathbf{D}_{test}$ contain 70% and 30% of the observations, respectively. The missing rate $r_{miss}$ is at 0.3. We construct two graphs $\mathcal{G}_{train}$ and $\mathcal{G}_{test}$ based on $\mathbf{D}_{train}$ and $\mathbf{D}_{test}$, respectively. We then train GRAPE with $\mathbf{D}_{train}$ and $\mathcal{G}_{train}$ using the same procedure as described in Section 4.2. At test time, we directly feed $\mathcal{G}_{test}$ to the trained GRAPE and evaluate its performance on predicting the missing values in $\mathbf{D}_{test}$. We repeat the same procedure for GAIN where training is also required. For all other baselines, since they do not need to be trained, we directly apply them to impute on $\mathbf{D}_{test}$.

**Results**. As shown in Figure 4, GRAPE yields 21% lower MAE compared with best baselines (MICE) without being retrained, indicating that our model generalizes seamlessly to unseen observations. Statistical methods have difficulties transferring the knowledge in the training data to new data. While GAIN is able to encode such information in the generator network, it lacks the ability to adapt to observations coming from a different distribution. However, by using a GNN, GRAPE is able to make predictions conditioning on the entire new datasets, and thus capture the distributional changes.

## 4.6 Ablation Study

**Edge dropout**. We test the influence of the edge dropout on the performance of GRAPE. We repeat the experiments in Section 4.2 for GRAPE with no edge dropout and the comparison results are shown in Section 4.6. The edge dropout reduces the test MAE by 33% on average, which verifies our assumption that using edge dropout could help the model learn to predict unseen edge values.

**Aggregation function**. We further investigate how the aggregation function (SUM($\cdot$), MAX($\cdot$), MEAN($\cdot$)) of GNN affects GRAPE's performance. While SUM($\cdot$) is theoretically most expressive, in our setting the degree of a specific node is determined by the number of missing values which is

Table 1: **Ablation study for GRAPE**. Averaged MAE of GRAPE on UCI datasets over 5 trials. Edge dropout (upper) reduces the average MAE by 33% on feature imputation tasks. MEAN(·) is adopted in our implementation. End-to-End training (lower) reduces the average MAE by 19% on prediction tasks (excluding two outliers).

|  | concrete | energy | housing | kin8nm | naval | power | protein | wine | yacht |
|---|---|---|---|---|---|---|---|---|---|
| Without edge dropout | 0.171 | 0.148 | 0.104 | 0.262 | 0.021 | 0.192 | 0.047 | 0.094 | 0.204 |
| **With edge dropout** | **0.090** | **0.136** | **0.075** | **0.249** | **0.008** | **0.102** | **0.027** | **0.063** | **0.151** |
| SUM(·) | 0.094 | 0.143 | 0.078 | 0.277 | 0.024 | 0.134 | 0.040 | 0.069 | 0.154 |
| MAX(·) | **0.088** | 0.142 | **0.074** | 0.252 | **0.006** | **0.102** | **0.024** | **0.063** | 0.153 |
| **MEAN(·)** | 0.090 | **0.136** | 0.075 | **0.249** | 0.008 | **0.102** | 0.027 | **0.063** | **0.151** |
| Impute then predict | 9.36 | 2.59 | 3.80 | 0.181 | **0.004** | 4.80 | 4.48 | **0.524** | 9.02 |
| **End-to-End** | **7.88** | **1.65** | **3.39** | **0.163** | 0.007 | **4.61** | **4.23** | 0.535 | **4.72** |

random and unrelated to the missing data task; in contrast, the MEAN(·) and MAX(·) aggregators are not affected by this inherent randomness of node degree, therefore they perform better.

**End-to-end downstream regression**. To show the benefits of using end-to-end training in label prediction, we repeat the experiments in Section 4.3 by first using GRAPE to impute the missing data and then perform linear regression on the imputed dataset for node labels (which is the same prediction model as the linear layer used by GRAPE). The results are shown in Section 4.6. The end-to-end training gets 19% less averaged MAE over all datasets except NAVAL and WINE. The reason for the two exceptions is similar as described in Section 4.3.

## 4.7 Further Discussions

**Scalability**. In our paper, we use UCI datasets as they are widely-used datasets for benchmarking imputation methods, with *both discrete and continuous features*. GRAPE can easily scale to datasets with thousands of features. We provide additional results on larger-scale benchmarks, including Flixster (2956 features), Douban (3000 features), and Yahoo (1363 features) in the Appendix. GRAPE can be modified to scale to even larger datasets. We can use scalable GNN implementations which have been successfully applied to graphs with billions of edges [55, 58]; when the number of features is prohibitively large, we can use a trainable embedding matrix to replace one-hot node features.

**Applicability of GRAPE**. In the paper, we adopt the most common evaluation regime used in missing data papers, i.e., features are missing completely at random. GRAPE can be easily applied to other missing data regimes where feature are not missing at random, since GRAPE is fully data-driven.

**More intuitions on why GRAPE works**. When a feature matrix does not have missing values, to make downstream label predictions, a reasonable solution will be directly feeding the feature matrix into an MLP. As is discussed in [57], an MLP can in fact be viewed as a GNN over a complete graph, where the message function is matrix multiplication. Under this interpretation, GRAPE extends a simple MLP by allowing it to operate on sparse graphs (*i.e.*, feature matrix with missing values), enabling it for missing feature imputation tasks, and adopting a more complex message computation as we have outlined in Algorithm 1.

## 5 Conclusion

In this work, we propose GRAPE, a framework to coherently understand and solve missing data problems using *graphs*. By formulating the *feature imputation* and *label prediction* tasks as edge-level and node-level predictions on the graph, we are able to train a Graph Neural Network to solve the tasks end-to-end. We further propose to adapt existing GNN structures to handle continuous edge values. Our model shows significant improvement in both tasks compared against state-of-the-art imputation approaches on nine standard UCI datasets. It also generalizes robustly to unseen data points and different data missing ratios. We hope our work will open up new directions on handling missing data problems with graphs.

## Broader Impact

The problem of missing data arises in almost all practical statistical analyses. The quality of the imputed data influences the reliability of the dataset itself as well as the success of the downstream tasks. Our research provides a new point of view for analysing and handling missing data problems with *graph representations*. There are many benefits to using this framework. First, different from many existing imputation methods which rely on good heuristics to ensure the performance [43], GRAPE formulates the problem in a natural way without the need of handcrafted features and heuristics. This makes our method ready to use for datasets coming from different domains. Second, similar to convolutional neural networks [24, 41], GRAPE is suitable to serve as a pre-processing module to be connected with downstream task-specific modules. GRAPE could either be pre-trained and fixed or concurrently learned with downstream modules. Third, GRAPE is general and flexible. There is little limitation on the architecture of the graph neural network as well as the imputation ($\mathbf{O}_{edge}$) and prediction ($\mathbf{O}_{node}$) module. Therefore, researchers can easily plug in domain-specific neural architectures, e.g., BERT [12], to the design of GRAPE. Overall, we see exciting opportunities for GRAPE to help researchers handle missing data and thus boost their research.

## Acknowledgments

We gratefully acknowledge the support of DARPA under Nos. FA865018C7880 (ASED), N660011924033 (MCS); ARO under Nos. W911NF-16-1-0342 (MURI), W911NF-16-1-0171 (DURIP); NSF under Nos. OAC-1835598 (CINES), OAC-1934578 (HDR), CCF-1918940 (Expeditions), IIS-2030477 (RAPID); Stanford Data Science Initiative, Wu Tsai Neurosciences Institute, Chan Zuckerberg Biohub, Amazon, Boeing, JPMorgan Chase, Docomo, Hitachi, JD.com, KDDI, NVIDIA, Dell. J. L. is a Chan Zuckerberg Biohub investigator.

## Footnotes

[1]Project website with data and code: `http://snap.stanford.edu/grape`

[1]We make data nodes and feature nodes to have the same feature dimension for the ease of implementation.

[1]Random forest is not included due to the lack of a public implementation that can handle missing data without imputation.

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
