[Supplementary Material]

# Appendix for "Handling Missing Data with Graph Representation Learning"

**Jiaxuan You**[1]* **Xiaobai Ma**[2]* **Daisy Yi Ding**[3]* **Mykel Kochenderfer**[2] **Jure Leskovec**[1]

[1]Department of Computer Science, [2]Department of Aeronautics and Astronautics,
and [3]Department of Biomedical Data Science, Stanford University
{jiaxuan, jure}@cs.stanford.edu
{maxiaoba, dingd, mykel}@stanford.edu

## A  Additional Details on Baseline Implementation

For imputation baselines including Mean, KNN, MICE, SVD, and Spectral, we use the implementation provided in the *fancyimpute* package[2]. For KNN, we use 50 nearest neighbors. For SVD, we set the *rank* equal to $m - 1$, where $m$ is the number of features. For MICE, we set the *maximum iteration number* to 3. For Spectral, we found the default heuristic for *shrinkage value* works the best. For a detailed explanation of the meaning of the parameters, we refer readers to the documentation of *fancyimpute* package. The hyper-parameter values are chosen by comparing the average imputation performance over all datasets. For GAIN, we use the source code released by the authors. All the hyper-parameters are the same as in the source code[3]. We use the *rpart* R package for the implementation of the decision tree method.

## B  Running Time Comparison

Here we report the running clock time for feature imputation of different methods at test time. For Mean, KNN, MICE, SAC, and Spectral, this means the running time of one function call for imputing the entire dataset. For GAIN and GRAPE, this means one forward pass of the network. Appendix B shows the averaged running time over 5 different trials with the same setting as described in Section 4.2.

Table 1: Running clock time (second) for feature imputation of different methods at test time.

|          | concrete | energy   | housing  | kin8nm  | naval   | power   | protein | wine    | yacht   |
|----------|----------|----------|----------|---------|---------|---------|---------|---------|---------|
| Mean     | 0.000806 | 0.000922 | 0.000942 | 0.00242 | 0.00596 | 0.00147 | 0.0127  | 0.00121 | 0.00064 |
| KNN      | 0.225    | 0.134    | 0.0913   | 9.95    | 30.1    | 11.4    | 656     | 0.504   | 0.0268  |
| MICE     | 0.0294   | 0.0311   | 0.0499   | 0.0749  | 0.256   | 0.0249  | 0.271   | 0.0531  | 0.027   |
| SVD      | 0.0659   | 0.0192   | 0.0359   | 0.162   | 0.0612  | 0.142   | 0.593   | 0.0564  | 0.0412  |
| Spectral | 0.0718   | 0.0565   | 0.0541   | 0.268   | 0.405   | 0.199   | 1.63    | 0.0978  | 0.0311  |
| GAIN     | 0.0119   | 0.0125   | 0.0131   | 0.017   | 0.0298  | 0.0146  | 0.0457  | 0.0131  | 0.0116  |
| GRAPE    | 0.0263   | 0.011    | 0.0115   | 0.0874  | 0.259   | 0.0488  | 0.568   | 0.0199  | 0.00438 |

## C  Comparisons with Additional Baselines

We additionally provide the comparison results of our method with two other state-of-the-art baselines: missMDA [2], a statistical multiple imputation approach, and MIWAE[3], a deep generative model.

We adapt the same setting as in Section 4.1 and the results are shown in Appendix C. GRAPE yields the smallest imputation error on all datasets compared with the two other baselines.

Table 2: Averaged MAE of *feature imputation* on UCI datasets at data missing level of 0.3.

|          | concrete | energy | housing | kin8nm | naval | power | protein | wine  | yacht |
|----------|----------|--------|---------|--------|-------|-------|---------|-------|-------|
| missMDA  | 0.190    | 0.225  | 0.142   | 0.285  | 0.038 | 0.215 | 0.068   | 0.090 | 0.226 |
| MIWAE    | 0.156    | 0.153  | 0.098   | 0.262  | 0.020 | 0.117 | 0.042   | 0.087 | 0.224 |
| GRAPE    | **0.090**| **0.136**| **0.075**| **0.249**| **0.008**| **0.102**| **0.027**| **0.063**| **0.151**|

## D   Experiments on Larger Datasets

To test the scalability of GRAPE, we perform additional *feature imputation* tests on the Flixter, Douban, and YahooMusic detests with preprocessed subsets and splits provided by [4]. The Flixster dataset has 2341 observations and 2956 features. The Douban dataset has 3000 observations and 3000 features. The YahooMusic dataset has 1357 observations and 1363 features. These datasets only have discrete values. We compare GRAPE with two GNN-based approaches, GC-MC [1] and IGMC [5]. The results are shown in Table 3, where the results of GC-MC and IGMC are provided by [5]. On all datasets, GRAPE shows a reasonable performance which is better than GC-MC and close to IGMC. Notice that the two baselines are specially designed for discrete matrix completion, where GRAPE is applicable to both continuous and discrete feature values and is general for both feature imputation and label prediction tasks.

Table 3: RMSE test results on Flixster, Douban, and YahooMusic.

|       | Flixster | Douban | Yahoo |
|-------|----------|--------|-------|
| GC-MC | 0.917    | 0.734  | 20.5  |
| IGMC  | **0.872**| **0.721**| **19.1**|
| Ours  | 0.899    | 0.733  | 19.4  |

## Footnotes

[2]https://github.com/iskandr/fancyimpute

[3]https://github.com/jsyoon0823/GAIN