[Reviews · NeurIPS 2020]

Review 1

Summary and Contributions: The article introduces a new framework for machine learning with missing values based on graph neural networks. The new method uses graph representations of the data, and allows to simultaneously predict missing features and learn downstream labels. The authors provide a simulation study comparing their method to existing feature imputation and label prediction techniques.

Strengths: - The paper provides a solution to an important issue in applied machine learning - The proposed method nicely unifies missing data imputation and subsequent learning tasks, without the need to combine imputation and prediction steps - The empirical results show significant improvement over existing techniques in terms of both missing values imputation and label prediction - The authors provided their code, which allows reproducibility, and a potential impact in the applied fields

Weaknesses: In my opinion, the main weaknesses of the paper are - the discussion of related work, which I believe is incomplete and sometimes incorrect (see section on relation to prior work) - the omission of important competitors in the numerical section, such as low-rank imputation methods for mixed data (e.g. https://www.jstatsoft.org/article/view/v070i01 available in R), deep latent variable models and autoencoders (e.g. reference [12] and http://proceedings.mlr.press/v97/mattei19a.html)

Correctness: The claims and methodology are sound

Clarity: The paper is very clear and well written

Relation to Prior Work: - There are several incorrect statements in the discussion concerning related works, namely: * l.25 matrix completion methods do allow for both categorical and continuous variables (e.g. https://arxiv.org/pdf/2005.12415.pdf, https://arxiv.org/abs/1806.09734, https://arxiv.org/pdf/1410.0342.pdf) * l.26 there exist matrix completion methods for online learning (e.g. https://arxiv.org/pdf/2002.08934.pdf, https://arxiv.org/abs/1906.07255) * l.35 Imputation approaches based on deep generative models can use feature values from other observations to impute missing values, e.g. by learning deep latent variable models (http://proceedings.mlr.press/v97/mattei19a/mattei19a.pdf) * l.37 Deep learning models for imputation can be initialized in many different ways. In addition, it is not clear to me how the proposed augmented node feature initialization solves this problem (is it also quite arbitrary).

Reproducibility: Yes

Additional Feedback: I read the other reviews as well as the authors feedback. Overall, I still advocate for acceptance, and I increased my score for the following reasons: - My main concerns were about the discussion of related work and additional experiments. In the rebuttal, the authors have satisfactorily answered them. - In particular, they have added experimental comparisons to two baselines (missMDA and MIWAE) and included the results in their response. They also added experiments on two additional high-dimensional data sets, which illustrates the scalability of the method.


Review 2

Summary and Contributions: This paper develops a graph representation learning method for handling missing data, where both missing features and labels can be estimated.

Strengths: (1) The paper is well written and the structure is very clear. (2) The novelty is good. This work has adopted a bipartite graph to build the connection between features, observations, and class labels, so that the missing data can be learned by the edge prediction and label can be estimated by node-level prediction. (3) The authors have also utilized a concatenation of the embedding or node or edge for handling the learning optimization.

Weaknesses: (1) The computational complexity should be analyzed. (2) The scalability on large-scale data should be discussed. (3) The numbers of features of all datasets used in the experiments are quite small, e.g. <500. The authors may explain the reason why using such data. (4) Compared to iterative SVD, there are many more competitive matrix completion methods, e.g.: [1] A. Goldberg, B. Recht, J. Xu, R. Nowak, and X. Zhu. Transduction with matrix completion: Three birds with one stone. NeurIPS 2010. [2] J. Fan and M. Udell. Online high rank matrix completion. CVPR 2019. One may like to know the advantage of the proposed method over these matrix completion methods.

Correctness: Yes.

Clarity: Yes.

Relation to Prior Work: Yes.

Reproducibility: Yes

Additional Feedback: I suggest the authors analyzing the computational complexity and discussing the advantage/disadvantage of the method compared to matrix completion algorithms such as [1][2]. The authors should give more explanation about the use of the datasets in the experiment. Is the algorithm applicable to vision data? It would be great if the authors could provide some theoretical guarantees for the proposed algorithm. %%% Thanks for the response and effort. I keep the score. One weakness of the work is that many advanced matrix completion methods were neglected. The "SVD" is an old and less effective one.


Review 3

Summary and Contributions: In this paper, the authors propose Grape, a framework for performing simultaneous missing data imputation and downstream tasks. The methodology relies on specifying a bipartite graph of data points and features, with edges specifying (known) feature values. Running a graph neural network over this graph allows for casting missing data imputation as link prediction and the downstream task as node-level prediction (e.g. classification or regression). Experiments are conducted on several UCI datasets, demonstrating favourable performance.

Strengths: Overall, the proposed framework has merit to it, is easily motivated, and could have potential for tackling missing data imputation in certain cases.

Weaknesses: The paper, unfortunately, has several potential issues concerning the novelty of the aspects of the proposed work and its actual applicability beyond the "UCI-like" domain that the authors are proposing here. I will outline the issues with the claims in the following section. Regarding scalability, there are many issues that will arise when deploying this idea on even moderately-sized datasets (with e.g. thousands of features). While the computation graph of GraphSAGE allows for explicit subsampling (which the authors don't seem to explicitly reference in the paper at all), there are still issues related to e.g. oversmoothing the message-passed signal at such scales (with a feature vertex having to handle potentially 1000s of incoming messages). This also poses an issue with the one-hot scheme used to encode feature nodes -- what if this becomes prohibitively large? The authors are invited to carefully consider, within their rebuttal and paper, whether and to what extent scalability represents an issue -- and how they would adapt their method to handle so. Ultimately, I would vote for rejecting the paper in its current form.

Correctness: Within the introduction and onwards throughout the paper, the authors describe several innovations of their work w.r.t. GraphSAGE, which overall don't appear to be clearly novel contributions to me: - Introducing edge embeddings is definitely not a new concept in GNNs and, in fact, predates GraphSAGE. Authors are invited to carefully compare their work to the MPNN of Gilmer et al. or GraphNets of Battaglia et al. - Augmented node features, especially the specific one-hot flavour introduced here, is also not a novel concept. There are many papers applying this trick ad-hoc to deal with transductive graph learning. One example of a paper that thoroughly analyses the impact this has to expressive power of GNNs is Relational Pooling (Murphy et al.), which proposes the relevant RP-GIN architecture. - Finally, the "edge dropout" technique is also a standard and well-known approach in graph representation learning. Besides DropEdge, that in my opinion the authors don't credit appropriately, to my knowledge this concept has been applied ad-hoc as early as the GAT paper (Veličković et al.). In all three cases, the authors need to make more careful comparisons of their proposed contributions to the existing literature on graph representation learning, and be more specific about their exact contributions with respect to related work.

Clarity: The paper is written well and easy to follow, with no issues in clarity.

Relation to Prior Work: Besides the above claims on correctness, the paper seems well-positioned within related work on feature imputation (though I'm not an expert in this area).

Reproducibility: Yes

Additional Feedback: I've read the other reviews as well as the authors' rebuttal. Thank you for your efforts! Giving authors the benefit of the doubt that they will accurately potray their GNN-related contribution in a subsequent iteration, and the experimental contributions of the rebuttal (which partially addresses some of my concerns), I will improve my score to weak reject (5). I am still not completely satisfied with the scalability aspect of the proposed architecture, however. Firstly, the authors claim "...the core of our method is a GNN, which has been successfully applied to graphs with billions of edge...", but this doesn't tell me how they will apply the GNN. Most of these successful applications relied on strong subsampling, which in this case would decimate many data points and/or features when making predictions. Secondly, "...we can use an embedding matrix to encode each feature, which has successfully handled millions of tokens in NLP..." -- I'm not convinced that this works from a representation learning perspective here. NLP applications thrived in presence of enormous datasets that allowed for these tokens to be repeatedly encountered. Here we will only have one graph representing our dataset, with few opportunities to transfer across datasets. Given the above, the authors' discussion does not completely convince me that prior successes in NLP and GNN applications will justify a successful scalable deployment here.


Review 4

Summary and Contributions: A method is presented for learning on data matrices with missing data by representing the data matrix as a bipartite graph with two node types-- one corresponding to observations and one to features. A GNN based on the GraphSAGE architecture is used to propagate messages through the graph. Edges between node types have a latent feature vector that is also updated. Feature imputation on the data matrix is therefore formulated as a linear function on this edge feature (more precisely, a function on the concatenation of the latent embedding of the incident nodes). Label prediction for examples is formulated as a linear function of example nodes. The formulation of the feature imputation and label prediction with missing data as a graph learning problem on a bipartite graph is not novel [1,2]. The main contribution is the architecture, which is fairly similar to GraphSAGE, with the addition of the edge features. [1] Berg, Rianne van den, Thomas N. Kipf, and Max Welling. "Graph convolutional matrix completion." arXiv preprint arXiv:1706.02263 (2017). [2] Zhang, Muhan, and Yixin Chen. "Inductive matrix completion based on graph neural networks." arXiv preprint arXiv:1904.12058 (2019).

Strengths: The formulation for handling missing data in a data matrix is well motivated and an important problem. In the domains considered, the bipartite graph formulation presented here performs well for both label prediction and feature imputation. The proposed architecture is intuitive for the bipartite graph formulation. Though intuitive, there is little concrete justification for the architecture choices as alternatives are not evaluated and other GNN work is not compared against in experiments

Weaknesses: The presented work is primarily an architecture development and empirical study rather than offering any theoretical developments. As such, it is limited by the experiments performed. (1) The architecture itself is not too deeply probed, for example comparing other graph embedding methods, ie alternative GNN architectures, to justify the architecture choices. Even existing work based on GNNs is not compared against [1,2 referenced in other sections of review] (2) Experiments are presented on 9 UCI datasets. It is not clear why these particular datasets were chosen. (3) The number of features for these datasets is small (reported between 6 and 9 features in line 163). (4) The missing data was simulated by randomly dropping features in the datasets, which may no reflect the distribution of missing data in real application where some features may be more likely to be missing for many observations or where some observations may be missing more features while other observations are fully observed, depending on the domain. -- (1) Authors have addressed in the rebuttal this by adding results for methods of [1,2] for comparison

Correctness: The methodology is well founded/explained and the empirical analysis seems sound, though incomplete since related GNN work is excluded.

Clarity: The writing is clear and the method developed and explained in a logical and clear manner.

Relation to Prior Work: Relevant related work for feature imputation and label prediction with missing data seem complete and well cited. However, other graph methods that use similar bipartite graph formulations [1,2] which are cited are not used as baselines. These seem the closest related works and should be included in experiments. The related works discussion only mentions that they make assumptions about the range of values that features take. It's not clear if these assumptions are valid in the domain that are considered in the experiments. Furthermore, in [2] a similar formulation is used (with a different embedding architecture) and feature imputation is performed by w^T \sigma(W concat(h_u,h_v)) for parameters w and W. This method which is closely related to the presented work seems to be easily applied to the problems considered here. Even label prediction could be performed with a simple modification of adding a linear function like O_node applied to h_u. [1] Berg, Rianne van den, Thomas N. Kipf, and Max Welling. "Graph convolutional matrix completion." arXiv preprint arXiv:1706.02263 (2017). [2] Zhang, Muhan, and Yixin Chen. "Inductive matrix completion based on graph neural networks." arXiv preprint arXiv:1904.12058 (2019). -- Authors have made an effort to include a discussion of these related works in the rebuttal to be included in the revised version

Reproducibility: Yes

Additional Feedback: Can authors clarify as I was not clear on this point: Are labels used at all in feature imputation or is the loss there only MSE on the edge value predicted by O_edge?. Labels, when available may be useful in imputing feature values just as the end-to-end training is shown to be useful in label prediction.

[Author Response · NeurIPS 2020]

We thank the reviewers for their constructive feedback. R1 and R2 appreciated the novelty of our unified framework that
solves both feature imputation and label prediction aspects for the missing data problem, and all reviewers agreed that
our solution is well motivated. Reviewers pointed out several main concerns, which we summarize and answer below:

**1 Novelty of the GNN method (R3 R4)**. We thank R3 and R4 for noticing that the core GNN components have been
separately used in other applications, and the formulation as a bipartite graph has been used for matrix completion task.
However, we emphasize that our main contribution is *not the particular GNN model but the graph-based framework*.
We show that a seemingly unrelated missing data problem (imputation and learning subsequent tasks) can naturally
be solved with graphs and we propose the first graph-based solution to it, which is acknowledged by R1 and R2.
Nevertheless, we agree with R3 and R4 that the justification
of the adopted method can be further improved and we will
do that in the final version of the paper. **New results to**
**justify our model.** In addition to ablation study in Section

| | concrete | energy | housing | kin8nm | naval | power | protein | wine | yacht |
|---|---|---|---|---|---|---|---|---|---|
| Sum | 0.094 | 0.143 | 0.078 | 0.277 | 0.024 | 0.134 | 0.040 | 0.069 | 0.154 |
| Max | **0.088** | 0.142 | **0.074** | 0.252 | **0.006** | **0.102** | **0.024** | **0.063** | 0.153 |
| **Mean** | 0.090 | **0.136** | 0.075 | **0.249** | 0.008 | **0.102** | 0.027 | **0.063** | **0.151** |

4.6, we further justify our model by showing how different types of aggregation, *i.e.*, MEAN(GraphSAGE-mean),
SUM(GIN), MAX(GraphSAGE-pool) affect the performance. These new results justify our selection of GraphSAGE-
style pooling in our architecture. **Justifying the new results.** While SUM is theoretically most expressive, in our setting
the degree of a specific node is determined by the number of missing values which is random and unrelated to the
missing data task; in contrast, the MEAN and MAX aggregators are not affected by this inherent randomness of node
degree. We will add these studies to justify our framework.

**2 Comparing with more baselines (R1 R2 R3)**. We thank reviewers for pointing out additional baselines to compare,
in addition to the 7 baselines that we have already examined (lines 169–181). **New baseline results.** Summarizing
reviewer's suggestions, we additionally compared our methods with 3 representative baselines on the feature imputation
task: (1) low-rank matrix completion method for mixed-type data (missMDA), (2) deep latent variable models and
autoencoders (MIWAE), (3) GNN-based baselines (GC-MC (Berg et al 2017) [1], IGMC (Zhang et al 2019) [2]).
**Discussion.** Note that *all the new baselines are only applicable to the feature imputation task* not the label prediction
task, and *the GNN-based baselines are only applicable to discrete-value data*, as they explicitly use different weights
for each distinct value, and thus cannot apply to mix-typed data; GRAPE *has none of these limitations*. We find that our
GRAPE framework can consistently outperform these new baselines in all the datasets. In 3 discrete-value datasets,
IGMC [2] has advantages as it is specifically designed for discrete-value data, while our model can still outperform
GC-MC [1]. We will include these new results and make sure to cite all the papers suggested in the reviews.

| | concrete | energy | housing | kin8nm | naval | power | protein | wine | yacht | | | Flixster | Douban | Yahoo |
|---|---|---|---|---|---|---|---|---|---|---|---|---|---|---|
| missMDA | 0.190 | 0.225 | 0.142 | 0.285 | 0.038 | 0.215 | 0.068 | 0.090 | 0.226 | | GC-MC [1] | 0.917 | 0.734 | 20.5 |
| MIWAE | 0.156 | 0.153 | 0.098 | 0.262 | 0.020 | 0.117 | 0.042 | 0.087 | 0.224 | | IGMC [2] | **0.872** | **0.721** | **19.1** |
| Ours | **0.090** | **0.136** | **0.075** | **0.249** | **0.008** | **0.102** | **0.027** | **0.063** | **0.151** | | Ours | 0.899 | 0.733 | 19.4 |

**3 Scalability (R2 R3 R4)**. We thank reviewers for asking to discuss the scalability of our method. We pick UCI
as they are widely-used datasets for benchmarking imputation methods, with *both discrete and continuous features*.
Our GRAPE *framework does not suffer from scalability issues*: the core of our method is a GNN, which has been
successfully applied to graphs with billions of edge; while R3 pointed out that a naive one-hot scheme will not work
when the number of features is prohibitively large, we can use an embedding matrix to encode each feature, which has
successfully handled millions of tokens in NLP. We agree with reviewers that experimenting on datasets with more
features can justify the scalability. We provide new results on large-scale benchmarks (Flixster: 2956 features, Douban:
3000 features, Yahoo: 1363 features) shown in the table above. Note that these datasets *only have discrete features*, and
therefore they were not used in our manuscript. We will include these new discussions and results in the revised paper.

**4 Discussions of related work (R1)**. We thank R1 for pointing out that our discussions for the related work can be
improved. We intended to give an overview for common limitations in lines 23–31 by starting with "often exhibit
notable shortcomings", but we do agree that this summary should be more rigorous. **Clarifications.** (1) We will cite
these matrix completion models designed for both discrete and continuous variables, and for online learning. (2) We
will change to "imputation approaches based on deep generative models do not *explicitly* use feature values from other
observations". (3) After formulating the missing data problem as a bipartite graph over feature and observation nodes,
the proposed augmented node feature initialization is natural: feature nodes are not permutation invariant so one-hot
tokens are used; observation nodes are permutation invariant so constant node features are used – we have provided a
thorough discussion in lines 130–149. Overall, we will include these discussions in the revised paper.

**5 Clarifications**. **Q: (R2)**"Applicable to vision data" **A:** Yes, *e.g.*, our model can be jointly used with CNNs and
trained end-to-end. **Q: (R2 R4)**"Theoretical guarantees" **A:** The success of our framework is supported by the universal
approximation capabilities of GNNs on graph structured data. **Q: (R4)** "Random missing data assumption" **A:** This is
the most common evaluation regime used in missing data papers. Our model can apply to other missing data scenarios.
**Q: (R4)** "Using labels in feature imputation" **A:** We agree that label information could be helpful in feature imputation
but we do not use it in our experiments, since the common evaluation for feature imputation tasks does not use labels.

[Meta-Review · NeurIPS 2020]

Dear authors, The reviewers discussed your document and carefully considered your rebuttal. All agree that the main contribution is the framework for dealing with missing values using bipartite graphs. This is an interesting idea, both for imputing missing values and for making predictions with missing values. They also appreciated that you added experimental comparisons to two reference methods (missMDA and MIWAE) and included the results in your response, as well as experiments on two additional high-dimensional data sets. Nevertheless, although they emphasized that GNNs are used here as a toolbox and not as the focus of the study, you need to be specific about important aspects of their application (such as discussions of architectural novelty and scalability), as noted by two reviewers. Yes, the reviewers are pleased to support acceptance of the paper. But they strongly encourage the authors to take seriously the issues related to their discussion of GNNs and to be thorough in how this technique is addressed in their camera-ready version. Therefore, as I also agree with them, I have decided to accept your paper, but you have to be very specific about scalability issues (and not just say that GNN in general has been applied to large graphics), there is no problem if this is also a problem for your case. In addition, in the introduction you need to position clearly your work with respects to the two works Graph Convolutional Matrix Completion and Inductive Matrix Completion Based on Graph Neural Networks as they also consider bibartite graph in the framework of missing values. In addition a few comments that you can try to integrate for the final version. Random forests dealing with missing values are available in the grf R package using the strategies for splitting missing incorporated in attributes MIA which is recommended when supervised learning with missing values (see the reference that may be useful https://arxiv.org/abs/1902.06931), also available in the partykit of the R packages. You can add the comparison when you first impute with mice or other methods and then when you predict with a non-parametric method such as a forest and not with a linear method. Finally, you should discuss the mechanism that generates the missing values (perhaps as a conclusion) as you only consider the MCAR scenario in the simulations and not MAR and MNAR.